# Genomic Investigation of Bacterial Co-Infection in Southern Pudu (*Pudu puda*) with Fatal Outcome: Application of Forensic Microbiology in Wildlife Impacted by Anthropogenic Disasters

**DOI:** 10.3390/ani15162435

**Published:** 2025-08-20

**Authors:** Valentina Aravena-Ramírez, Edhnita Inostroza-Muñoz, Fredy Riquelme, César Mellado, Nilton Lincopan, Paula Aravena, Danny Fuentes-Castillo

**Affiliations:** 1Departamento de Patología y Medicina Preventiva, Facultad de Ciencias Veterinarias, Universidad de Concepción, Chillán 3780000, Chile; varavena2018@udec.cl (V.A.-R.); edhnitasoleinos@udec.cl (E.I.-M.); fredriquelme@udec.cl (F.R.); cemelladob@udec.cl (C.M.); 2Centro de Rehabilitación de Fauna Silvestre, Universidad de Concepción, Chillán 3780000, Chile; paularavena@udec.cl; 3Department of Microbiology, Instituto de Ciências Biomédicas, Universidade de São Paulo, São Paulo 05508-020, Brazil; lincopan@usp.br; 4Department of Clinical Analysis, School of Pharmacy, Universidade de São Paulo, São Paulo 05508-070, Brazil; 5Department of Pathology, School of Veterinary Medicine and Animal Science, Universidade de São Paulo, São Paulo 05508-070, Brazil; 6Departamento de Ciencia Animal, Facultad de Ciencias Veterinarias, Universidad de Concepción, Chillán 3780000, Chile

**Keywords:** ESBL, wildlife conservation, *Escherichia coli* ST224

## Abstract

The southern pudu is a vulnerable deer species impacted by anthropogenic threats and infectious pathogens. In this context, we investigated bacterial infections affecting a southern pudu admitted to a wildlife rehabilitation center after suffering dog bites and limb burns caused by wildfires. Utilizing genomic tools, we characterized a triple bacterial infection comprising *Escherichia coli* ST224, *Klebsiella oxytoca* ST145, and *Acinetobacter baumannii* ST1365. Due to its broad resistome and virulome, a clone of *E. coli* ST224 progressed from a soft tissue infection to fatal sepsis. This work highlights the significant utility and accuracy of genomic research and forensic microbiology in understanding the infectious disease process that threatens wild animals.

## 1. Introduction

Rehabilitating wildlife species and understanding the effects of pathogens and infectious diseases are crucial aspects of veterinary medicine. These efforts improve the health and survival of wildlife and safeguard human populations from zoonotic threats [1]. Infectious diseases pose a significant threat to the conservation of wild species, such as the southern pudu (*Pudu puda*), one of the smallest deer species in the world. Their conservation status is vulnerable, with declining populations mainly due to anthropogenic effects, including roadkill, forest fires, and free-roaming dog attacks [2]. There is molecular and serological evidence of the presence of potential pathogens in the southern pudu [3,4,5,6]. Therefore, the only documented case linking specific infectious agents to disease in this species is the association of mycotic pneumonia caused by *Aspergillus fumigatus* or *Mucor* spp. and encephalitis caused by *Curvularia spicifera* in zoo-captive southern pudus [7].

Forensic microbiology employs microbiological methods for analyzing evidence of criminal cases, bioterrorism, outbreaks, and the transmission of pathogens [8,9]. In wildlife, this little-explored science has emerged as a valuable tool for tracing the origin and spread of infectious diseases and for investigating international wildlife trafficking [10,11,12]. For a most effective microbial forensic analysis, sufficient basic scientific information concerning microbial genetics, evolution, physiology, and ecology is required [13]. In this sense, whole-genome sequencing (WGS) is a valuable and versatile tool, and its use in clinical settings has been proposed to investigate the genomics of infectious diseases [14]. Its application in the forensic field is noteworthy, as it allows genomic epidemiology and comparative microbial genomics, identifying the origin and potential spread of infectious pathogens [15].

This study aims to analyze the genomic characteristics of infection-causing bacterial pathogens and investigate the cause of death of a specimen of *P. puda* admitted to a wildlife rehabilitation center in Chile.

## 2. Materials and Methods

### 2.1. Case Background and Sample Collection

The forest fires that occurred in south-central Chile during the summer of 2023 were primarily ascribed to anthropogenic factors. Official data from the Chilean government indicates that none of the fires recorded during the 2022–2023 season were caused by natural factors [16,17]. In this context, the wildlife rehabilitation center at the University of Concepcion received an adult male southern pudu from Florida, Biobio region, Chile. The patient had wounds in the lumbosacral area attributed to dog bites (as declared by their rescuers) and a burned right hind limb, both of which showed signs of infection (Appendix A). The deer received empiric fluoroquinolone (i.e., enrofloxacin, 5 mg/kg, IM, q. 24 h), with unsuccessful results. Swabs from both injuries were taken for microbiological culture. The patient passed away thirteen days after being admitted. Considering the medical records of bacterial infections in the patient and the absence of an established cause of death, an intracardiac blood sample was collected for microbiological analysis at necropsy. Additionally, an oval nodule measuring 70 × 100 mm was identified in the abdominal cavity. It had a smooth external surface, firm consistency, and calcified yellowish-white content, which was also sampled (Appendix A).

### 2.2. Isolation and Bacterial Identification

For samples from wounds and the internal abdominal nodule, microbiological cultures were performed using brain–heart agar, blood agar, and MacConkey agar, and incubated for 24 h at 37 °C. For an intracardiac blood sample, the culture was initially grown in brain–heart infusion broth for 24 h at 37 °C, and subsequently transferred to brain–heart agar, blood agar, and MacConkey agar for an additional 24 h incubation at 37 °C. An oxidase test for the detection of cytochrome oxidase in microorganisms was performed by using Bactident^®^ (Sigma-Aldrich, St. Louis, MO, USA), according to the manufacturer’s instructions. Preliminary bacterial identification was performed through API^®^ 20E™ (for oxidase-negative isolates) or API^®^ 20NE™ (for oxidase-positive isolates) systems (BioMérieux, Marcy-l’Étoile, France). For the susceptibility testing, the selection of antibiotics varied according to the analyzed bacteria, including cefazolin, cefovecin, ceftriaxone, cefoperazone, ceftazidime, cefoxitin, cefoxitin, cefotaxime, cefepime, amoxicillin/clavulanate, piperacillin/tazobactam, imipenem, meropenem, tetracycline, gentamicin, amikacin, chloramphenicol, trimethoprim–sulfamethoxazole, enrofloxacin, and levofloxacin. Susceptibility interpretation followed the protocols and cutoff points established by the Clinical and Laboratory Standards Institute [18,19]. *Escherichia coli* ATCC 25922 was used as a control strain.

### 2.3. WGS and Genomic Characterization

Total genomic DNA of the bacterial isolates was extracted using the InstaGene™ Matrix (Bio-Rad Laboratories, Hercules, CA, USA) extraction method and subjected to WGS using the Illumina NextSeq 2000 platform (Illumina, Inc., San Diego, CA, USA). Genomic assembly was performed using the Shovill (Version 1.1.0) with the SKESA assembler, and the quality control and characterization of the genomes were performed using the CheckM lineage_wf and Quast (Appendix A), tools available in the Galaxy web-based platform https://usegalaxy.org/ (accessed on 12 May 2025). For *E. coli*, multilocus sequence type (MLST), resistome, serotype prediction, and plasmid replicons were identified using MLST 2.0, ResFinder 4.1.0, SerotypeFinder 2.0, and PlasmidFinder 2.1, respectively, available from the Center for Genomic Epidemiology (http://genomicepidemiology.org/, accessed on 12 May 2025). For the virulome analysis, the VFDB database (https://www.mgc.ac.cn/VFs/, accessed on 12 May 2025) was used. For *Klebsiella oxytoca*, the MLST, resistome, virulome, and plasmid replicons were analyzed using the same databases as those for *E. coli.* For *Acinetobacter baumannii*, the Pasteur scheme was used for MLST analysis available in the PubMLST database (https://pubmlst.org/organisms/acinetobacter-baumannii, accessed on 12 May 2025); while resistome and virulome were obtained from ResFinder 4.1.0 and VFDB (2025), respectively. The *A. baumannii* capsule was also typed using the Kaptive online tool (https://kaptive-web.erc.monash.edu/, accessed on 12 May 2025) to predict serotypes (K-type and O-type). For all strains, mutations in the quinolone resistance-determining regions (QRDRs) were investigated using the Resistance Gene Identifier 6.0.5 (RGI) tool from the CARD 4.0.1 database (https://card.mcmaster.ca/home, accessed on 12 May 2025). For all predicted resistance genes, a ≥97% identity/coverage threshold was used as a filter for identification. For virulome and plasmid replicons, the default results were considered for each database. The raw data is available at the National Center for Biotechnology Information (NCBI) under the BioProject accession number PRJNA1269607.

### 2.4. Phylogenetic and Clonality Analysis

To elucidate the phylogenetic relationship of the two *E. coli* ST224, the genomic sequences of our isolates were deposited in the Enterobase database (https://enterobase.warwick.ac.uk/, accessed on 12 May 2025). Subsequently, a phylogenomic analysis was conducted using the cgMLST V1 + HierCC V1 scheme and MSTree V2 algorithms. This analysis included 737 available genome sequences of *E. coli* ST224, containing information regarding the sample origin, country, and collection year (Appendix A). To determine clonality among strains, the clade including our *E. coli* ST224 strains was subjected to single-nucleotide polymorphism (SNP) analysis using the CSI Phylogeny 1.4 platform of the Center for Genomic Epidemiology, and clonality was interpreted with thresholds following previously described guidelines [20]. Closely related genome assemblies (fewer than 100 SNP differences) were selected to construct the final SNP-based phylogenetic tree (Appendix A). The resistome of the closely related genome assemblies was obtained using the databases previously utilized with *E. coli* to compare genomic and epidemiological data.

## 3. Results

### 3.1. Bacterial Isolation and Antimicrobial Susceptibility

From infected wounds in the lumbosacral area attributed to dog bites, a *K. oxytoca* MVL-12-23 strain was isolated. From burn wounds on the extremities and intracardiac blood samples, *E. coli* MVL-11-23 and MVL-123-23 strains were isolated, respectively. Finally, from the oval nodule detected at the necropsy, the *A. baumannii* MVL-13-23 strain was detected. The *K. oxytoca* MVL-12-23 strain exhibits phenotypic resistance to cefazolin, tetracycline, gentamicin, chloramphenicol, trimethoprim–sulfamethoxazole, and enrofloxacin. The two *E. coli* MVL-11-23 and MVL-123-23 strains displayed resistance to cefazolin, cefovecin, ceftriaxone, gentamicin, trimethoprim–sulfamethoxazole, enrofloxacin, and intermediate resistance to cefoperazone, amoxicillin/clavulanate, and amikacin. The *A. baumannii* MVL-13-23 strain exhibits intermediate resistance to cefotaxime (Table 1).

### 3.2. Genomic Characterization of Infection-Causing Bacteria in Southern Pudu

The *K. oxytoca* MVL-12-23 strain belonged to the ST145 lineage, carrying resistance determinant genes against beta-lactams (*bla*_OXY-2-10_), aminoglycosides [*aadA1*, *aadA5*, *aac(3)-Iia*], macrolides [*mph(A)*], phenicols (*catA1*), tetracyclines [*tet(B)*], sulfonamides (*sul1*), and trimethoprim (*dfrA17*). In addition, it has mutations in the *gyrA* (S83I) and *gyrB* (S463A) QRDR, associated with fluoroquinolone resistance (Table 2). The virulome comprised genes conferring bacterial adherence, iron uptake, secretion systems, efflux pumps, nutritional factor, virulence regulation, cell surface components, magnesium uptake, protease, and stress adaptation (Appendix A). The plasmid replicons detected were *IncFIB(K)* and *IncM1* (Table 2).

The two *E. coli* strains belonged to the same ST224 lineage. The resistome was composed of genes conferring resistance to beta-lactams (*bla*_CTX-M-1_), aminoglycosides [*aac(3)-IId*, *aph(3′)-Ia*, *aph(3″)-Ib*, *aph(6)-Id*], macrolides [*mph(A)*], sulfonamides (*sul2*), and trimethoprim (*dfrA17*). In addition, the *E. coli* strains displayed mutations in the QRDR *gyrA* (D87N and S83L) and *parC* (S80I) genes, which confer resistance to fluoroquinolones. The serotype prediction of both *E. coli* strains was O126:H23 (Table 2).

For the *E. coli* MVL-11-23 strain, the virulome includes virulence factors associated with adherence, invasion, iron uptake, toxin, autotransporter, non-lee-encoded ttss effectors, secretion system, and *Yersinia* O antigen (Appendix A). On the other hand, the *E. coli* MVL-123-23 strain carried a virulome composed of genes encoding for adherence, invasion, iron uptake, secretion systems, toxins, endotoxin, serum resistance, immune evasion, antiphagocytosis, magnesium uptake, quorum sensing, motility, autotransporter, non-lee-encoded ttss effectors, virulence regulation, amino acid and purine metabolism, anaerobic respiration, cell surface components, chemotaxis and motility, efflux pump, enolase enzyme, lipid and fatty acid metabolism, nutritional virulence, stress adaptation, acyltransferases, and *Yersinia* O antigen (Appendix A).

The plasmid replicons detected were *IncM1*, *p0111*, and IncQ1 for the *E. coli* MVL-11-23 strain and *p0111* and IncQ1 for the *E. coli* MVL-123-23 strain.

The *A. baumannii* MVL-13-23 strain belonged to ST1365 and serotype KL138, OCL1. It harbored a resistance gene *bla*_OXA-413_ that confers resistance to beta-lactams, as well as mutations in *parC* (V104I and D105E) QRDR. The virulome comprises genes that confer characteristics such as bacterial adherence, biofilm formation, iron uptake, quorum sensing, phospholipases, immune evasion, serum resistance, sensor kinases, and catalase (Appendix A).

### 3.3. Phylogenetic and Clonality Analysis of E. coli ST224

The initial phylogeny of *E. coli* ST224 was conducted with 737 genome assemblies from different countries around the world that met the established criteria (host, country, and year of collection data) (Appendix A). The two *E. coli* strains isolated from *P. puda* were closely related (<100 SNPs of difference) to strains from Brazil (wild bird), Switzerland (human, cat, dog, and house environment), and the United States (human, pig, horse, and dog) (Appendix A).

The collection years ranged from 2019 to 2024 (Figure 1). The resistome comparison of the strains includes resistance genes to beta-lactams (*bla*_NDM-5_, *bla*_CTX-M-1_), aminoglycosides [*aac(3)-IId*, *aph(3′)-Ia*, *aph(3″)-Ib*, *aph(6″)-Ib3*, *aph(6)-Id*], macrolides [*mph(A)*, *mph(E)*, *msr(E)*], tetracyclines [*tet(A)*], sulfamethoxazole (*sul2)* trimethoprim *(dfrA17)*, phenicols (*catA2*), and rifampicin (*ARR-3*) antibiotics (Figure 1). The *E. coli* MVL-11-23 strain isolated from the infected wound differed by five SNPs from the *E. coli* MVL-123-23 strain isolated from the cardiac blood sample.

## 4. Discussion

This study investigated the genomic characteristics of a triple bacterial co-infection in a vulnerable *P. puda* impacted by anthropogenic activities with a fatal outcome, using WGS as a relevant tool in the forensic field.

Wildfires result in the loss of millions of hectares annually due to uncontrolled blazes, severely impacting the environment, wildlife, and human life [21]. These disasters have a profound impact on biodiversity, leading to habitat loss, reductions in the population sizes of both flora and fauna, alterations in ecosystems, and environmental pollution [22,23,24]. While wildlife has evolved an escape response to fire, this does not guarantee survival in the face of such events [25,26,27]. As in the case described in this paper, disoriented escaped animals can be seen as prey of domestic or wild carnivores. On the other hand, wildlife that survives wildfires may be directly impacted by secondary infections due to wound contamination, which reduces the chances of survival for the burned animals [28].

*E. coli* is a diverse bacterial species comprising both commensal and pathogenic strains capable of causing intestinal and extraintestinal diseases in humans and animals. Advances in genomics have revealed that acquiring virulence-associated genes through horizontal gene transfer plays a key role in its pathogenic potential [29]. Detailed genomic analysis was prioritized for the two *E. coli* ST224 isolates to confirm clonal dissemination from a wound to intracardiac blood, indicating progression to septicemia. This clone also carried an extensive set of virulence factors and critical resistance determinants, features not observed in *K. oxytoca* ST145 (MVL-12-23) or *A. baumannii* ST1365 (MVL-13-23), which were restricted to localized lesions without evidence of systemic spread. In this context, the southern pudu suffered from a secondary bacterial infection in its burned right hind limb caused by a multidrug-resistant CTX-M-1-producing *E. coli* ST224, progressing from local soft tissue infection to fatal sepsis. This was confirmed by SNP and virulome analyses, which verified that it was an *E. coli* ST224 clone identified in the bloodstream sample (with five SNPs of difference) [20]. However, the strain recovered from blood carried a more extensive virulome, which may have contributed to an increased pathogenic potential. Detected genes that encode for bacterial invasion, iron uptake, hemolysins, antiphagocytosis, chemotaxis and motility, endotoxin, immune evasion, quorum sensing, serum resistance, and stress adaptation could have facilitated the septicemia and fatal outcome in the case [30,31]. Unfortunately, we cannot confirm the specific mobile genetic elements (MGEs) carrying this extra virulome due to the short-read sequencing that does not allow a correct assembly of plasmids or other MGEs [32]. The marked differences in virulome composition between the two *E. coli* ST224 isolates, despite their clonal relationship based on core genome SNP analysis, suggest the presence of distinct subpopulations within the same lineage [33]. Given the high assembly quality and the absence of mobile genetic elements typically linked to large-scale virulence gene acquisition, it is unlikely that these differences arose solely during the course of infection. However, the possibility of virulence gene acquisition through bacterial transformation cannot be entirely excluded, although no direct evidence supports this event in the present case. These findings highlight the genomic plasticity within pathogenic lineages and the potential coexistence of subpopulations with variable virulence potential in the same host. On the other hand, limited therapeutic options due to the antimicrobial resistance determinants, including the production of the extended-spectrum beta-lactamase (ESBL) CTX-M-1 by the strain, contributed to the death of the animal. ESBL-producing Enterobacterales are classified within the critical priority group of the WHO list, and the presence of this type of enzyme produced by *E. coli* complicates the treatment of patients [34]. Identifying *E. coli* ST224 lineage in the southern pudu adds to those reported worldwide in diverse hosts such as humans, pets, livestock, wildlife, and the environment [35]. This demonstrates that this One Health lineage adapts to different species and hospital and wild environments.

In addition to the *E. coli* infection, the patient had an infection in a dog bite wound where a multidrug-resistant *K. oxytoca* ST145 was isolated, being the first report in wildlife. One limitation of this study regarding the wounds in the lumbosacral region is that we could not confirm whether they were caused by attacks from domestic dogs, as reported by the rescuers, or by other wild canids, such as wild foxes. The *K. oxytoca* ST145 lineage has been described in Poland, China, and Spain as an emerging pathogen primarily causing nosocomial post-surgical or wound infections in humans [36,37,38,39]. This increases the need to monitor the environment of wildlife rehabilitation centers and wild patients with secondary wound infections. Both *E. coli* ST224 and *K. oxytoca* ST145 were resistant to fluoroquinolones due to point mutations in quinolone resistance-determining regions, leading to the therapeutic failure with the empiric enrofloxacin administered.

Finally, *A. baumannii* ST1365 was a pathological finding in the southern pudu, establishing the first confirmed report of this bacterium by WGS in wildlife. While not characteristic, such presentations highlight the potential of *A. baumannii* to induce focal, nodular pathology under certain clinical conditions [40,41]. The duration and underlying causes of the lesion caused by *A. baumannii* in the southern pudu remain unknown. Although this pathogen is well studied in human medicine, its pathogenic potential in animals warrants further investigation [42,43]. Both *K. oxytoca* and *A. baumannii* are recognized opportunistic pathogens with the potential to cause disease in humans and animals. In this case, they were associated only with localized lesions and showed no evidence of systemic dissemination, suggesting that their role was limited to secondary infections rather than primary drivers of the fatal outcome.

## 5. Conclusions

This study highlights the utility of WGS as a powerful tool for forensic microbiology in wildlife, enabling the precise characterization of pathogens responsible for fatal infections. The investigation of a deceased southern pudu revealed a complex case of a triple bacterial infection involving *E. coli* ST224, *K. oxytoca* ST145, and *A. baumannii* ST1365, exhibiting significant antimicrobial resistance and extensive virulence gene repertoires. The detection of CTX-M-1-producing *E. coli* ST224 emphasizes the One Health implications of multidrug-resistant bacteria circulating across human, animal, and environmental reservoirs. This study suggests that a clonally related *E. coli* ST224, initially isolated from a wound infection, could cause fatal septicemia in the southern pudu, underscoring the pathogen’s adaptability and virulent potential in wildlife hosts. The findings also underscore the underexplored role of opportunistic pathogens such as *A. baumannii* and *K. oxytoca* in wildlife health. This case study exemplifies the importance of implementing genomic epidemiology in wildlife rehabilitation settings to detect, monitor, and understand the emergence and spread of infectious diseases, ultimately contributing to wildlife conservation efforts and safeguarding public health.

## Figures and Tables

**Figure 1 animals-15-02435-f001:**
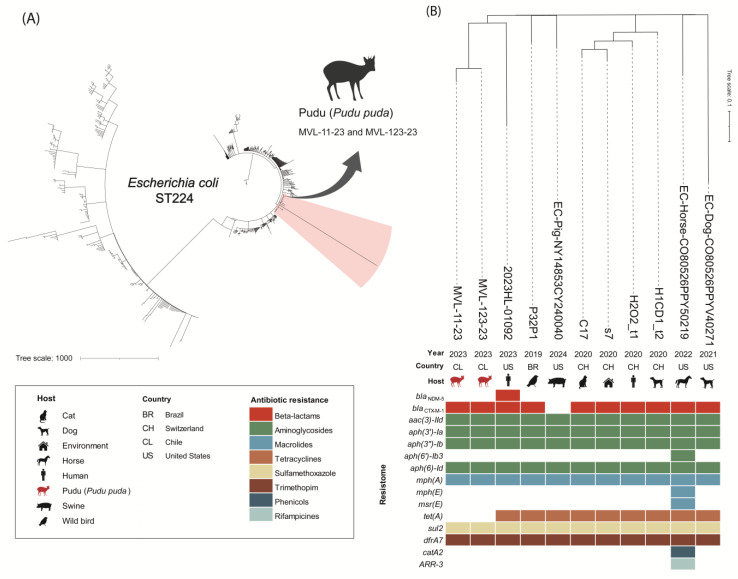
Phylogenomic analysis and comparison of *Escherichia coli* ST224 from infected wound and intracardiac blood samples isolated from southern pudu and closely related genomic sequences available in the Enterobase database. (**A**) Phylogenomic analysis of 737 genomes of *E. coli* ST224, highlighting the clade that includes strains isolated from the southern pudu. (**B**) Detailed view of the highlighted clade containing *E. coli* ST224 from southern pudu and closely related genomic sequences (fewer than 100 SNPs of difference) with comparison of epidemiological data and resistome.

**Table 1 animals-15-02435-t001:** Information and antimicrobial susceptibility profile of infection-producing bacteria in southern pudu.

Strain	Bacterial Identification	Source	Antibiotic Resistance Profile ^c^		
			R	I	S
MVL-012-23	*Klebsiella oxytoca* ^a^	Wounds in the lumbosacral area	KZ, TE, CN, AK, C, SXT, ENR	-	CRO, CFP, CAZ, FOX, AMC, IMP, MEM, AK
MVL-011-23	*Escherichia coli* ^a^	Infected burn wounds	CRO, CVN, KZ, CN, SXT, ENR	CFP, AMC, AK	CAZ, FOX, IMP, MEM, C
MVL-123-23	*Escherichia coli* ^a^	Intracardiac blood	CRO, CVN, KZ, CN, SXT, ENR	CFP, AMC, AK	CAZ, FOX, IMP, MEM, C
MVL-013-23	*Acinetobacter baumannii*/*calcoaceticus* ^b^	Internal abdominal nodule	-	CTX	CAZ, FEP, PTZ, IMP, MEM, TE, CN, AK, SXT, LEV

^a^ Bacterial identification by using API 20E™; ^b^ bacterial identification by using API+ 20NE™; ^c^ R, resistant; I, intermediate; S, susceptible. KZ, cefazolin; CVN, cefovecin; CRO, ceftriaxone; CFP, cefoperazone; CAZ, ceftazidime; FOX, cefoxitin; CTX, cefotaxime; FEP, cefepime; AMC, amoxicillin/clavulanate; PTZ, piperacillin/tazobactam; IMP, imipenem; MEM, meropenem; TE, tetracycline; CN, gentamicin; AK, amikacin; C, chloramphenicol; SXT, trimethoprim–sulfamethoxazole; ENR, enrofloxacin; LEV, levofloxacin.

**Table 2 animals-15-02435-t002:** Multilocus sequence typing (MLST), resistome, serotype prediction, and plasmid replicons of *Klebsiella oxytoca*, *Escherichia coli*, and *Acinetobacter baumannii* strains detected in southern pudu.

Bacterial Strain	MLST	Resistome ^a^	Serotype Prediction	Plasmid Replicons
*K. oxytoca* MVL-12-23	ST145	*bla_OXY-2-10_* ^1^, *aadA1* ^2^, *aadA5* ^2^, *aac(3)-IIa* ^2^, *mph(A)* ^3^, *tet(B)* ^4^, *sul1* ^5^, *dfrA17* ^6^, *catA1* ^7^, *gyrA (S83I)* ^8^, *gyrB (S463A)* ^8^	ND	*IncFIB(K)*, *IncM1*
*E. coli* MVL-11-23	ST224	*bla*_CTX-M-1_ ^1^, *aac(3)-IId* ^2^, *aph(3′)-Ia* ^2^, *aph(3″)-Ib* ^2^, *aph(6)-Id* ^2^, *mph(A)* ^3^, *sul2* ^5^, *dfrA17* ^6^, *gyrA* (D87N and S83L), *parC* (S80I)	O126:H23	*IncM1*, *IncQ1*, *p0111*
*E. coli* MVL-123-23	ST224	*bla*_CTX-M-1_ ^1^, *aac(3)-IId* ^2^, *aph(3′)-Ia* ^2^, *aph(3″)-Ib* ^2^, *aph(6)-Id* ^2^, *mph(A)* ^3^, *sul2* ^5^, *dfrA17* ^6^, *gyrA* (D87N and S83L) ^8^, *parC* (S80I) ^8^	O126:H23	*IncQ1*, *p0111*
*A. baumannii* MVL-13-23	ST1365	*bla*_OXA-413_ ^1^, *parC* (V104I, and D105E) ^8^	KL138, OCL1	-

^a^, genes encoding resistance to the following: ^1^, beta-lactams; ^2^, aminoglycosides; ^3^, macrolides; ^4^, tetracyclines; ^5^, sulfamethoxazole; ^6^, trimethoprim; ^7^, phenicols; ^8^, fluoroquinolones. ND, not determined.

## Data Availability

The raw data is available at the National Center for Biotechnology Information (NCBI) under the BioProject accession number PRJNA1269607.

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
