# Peer review of "Genomic Investigation of Bacterial Co-Infection in Southern Pudu (Pudu puda) with Fatal Outcome: Application of Forensic Microbiology in Wildlife Impacted by Anthropogenic Disasters"

_animals, 2025, doi:10.3390/ani15162435_

Round 1

Reviewer 1 Report

Comments and Suggestions for Authors

The authors presented a relevant and interesting work about the use of genomics and microbiology in a context of defining death causes of a threatened species (Pudu puda).

However, I believe there is a little bit of lack of context in two single aspects than can be easily approached by the authors. 

1) Introduction - improve your introduction with some aspects regarding forensic sciences impact in willdife conservations to have a better context about the use of forensic microbiology. Authors say " Its application in the forensic field is noteworthy, as it 
allows genomic epidemiology and microbial genomics, identifying the origin and poten
tial spread of infectious pathogens". Yes, but what about the role of forensic microbiology (or forensic entomology or other forensic sciences) in willdife conservation, animal trade etc... I believe it would be important to give that perspective, otherwise we cannot actually say that your work belong to the field of forensic microbiology, being just a genomic investigation of the death cause of this animal, as a simple clinical case. 

2) I also have similar concern regarding the attribution of this death to anthropogenic causes. Of course, burns due to forest fire or dog bites are recognizable by experienced vets working in willdife rescue centers but if you have photos to illustrate this, it would significantly improve the manuscript. Forensic sciences are very dependent on necropsy reports and photos and I believe some context regarding the situation of fires in the region (because fires can have human or natural causes) as well as the presence of dogs (because bites could come from other carnivores) is necessary. Basically, authors should answer to the question "why do you believe these lesions represent the anthropogenic threat(s )that this species is facing in the area?". I believe in can be answered in the Discussion, but providing some context first on Introduction or on a first methodology subchapter is also a possibility. 

Author Response

1. Summary

We sincerely thank the reviewers for their valuable time, thoughtful comments, and constructive suggestions, which have greatly improved the quality of our manuscript. Please find below our detailed responses to each comment. All corresponding changes have been incorporated into the revised manuscript and are highlighted using track changes.

2. Questions for General Evaluation

Reviewer’s Evaluation

Response and Revisions

Does the introduction provide sufficient background and include all relevant references?

Can be improved

We respond point by point below.

3. Point-by-point response to Comments and Suggestions for Authors

Comments 1: Introduction - improve your introduction with some aspects regarding forensic sciences impact in willdife conservations to have a better context about the use of forensic microbiology. Authors say " Its application in the forensic field is noteworthy, as it allows genomic epidemiology and microbial genomics, identifying the origin and potential spread of infectious pathogens". Yes, but what about the role of forensic microbiology (or forensic entomology or other forensic sciences) in willdife conservation, animal trade etc... I believe it would be important to give that perspective, otherwise we cannot actually say that your work belong to the field of forensic microbiology, being just a genomic investigation of the death cause of this animal, as a simple clinical case. 

Response 1: We appreciate your observation and fully agree with your comment. Due to the lack of information supporting forensic microbiology in the text, we have elaborated further on this topic in the introduction, as recommended.

Lines 60 – 70: Forensic microbiology employs microbiological methods for analyzing evidence of criminal cases, bioterrorism, outbreaks and transmission of pathogens [8,9]. In wildlife, this little explored science has emerged as a valuable tool for tracing the origin and spread of infectious diseases and for investigating international wildlife trafficking [10-12]. For a most effective microbial forensic analysis, sufficient basic scientific information concerning microbial genetics, evolution, physiology, and ecology is required [13]. In this sence, the whole genome sequencing (WGS) is a valuable and versatile tool, and its use in clinical settings has been proposed to investigate the genomics of infectious diseases [14]. Its application in the forensic field is noteworthy, as it allows genomic epidemiology and comparative microbial genomics, identifying the origin and potential spread of infectious pathogens [15].

Additional references:

[10] Foster, G., Whatmore, A. M., Dagleish, M. P., Malnick, H., Gilbert, M. J., Begeman, L., Macgregor, S. K., Davison, N. J., Roest, H. J., Jepson, P., Howie, F., Muchowski, J., Brownlow, A. C., Wagenaar, J. A., Kik, M. J. L., Deaville, R., Doeschate, M. T. I. ten, Barley, J., Hunter, L., & IJsseldijk, L. L. (2019). Forensic microbiology reveals that Neisseria animaloris infections in harbour porpoises follow traumatic injuries by grey seals. Scientific Reports, 9(1), 14338. https://doi.org/10.1038/s41598-019-50979-3

[13] American Society for Microbiology. (2003). Microbial Forensics: A Scientific Assessment: This report is based on a colloquium sponsored by the American Academy of Microbiology held June 7-9, 2002, in Burlington, Vermont. In American Academy of Microbiology Colloquia Reports. https://doi.org/10.1128/AAMCOL.7JUNE.2002

[9] Oliveira, M., & Amorim, A. (2018). Microbial forensics: new breakthroughs and future prospects. Applied Microbiology and Biotechnology, 102(24), 10377–10391. https://doi.org/10.1007/s00253-018-9414-6

[11] Smart, U., Cihlar, J. C., & Budowle, B. (2021). International Wildlife Trafficking: A perspective on the challenges and potential forensic genetics solutions. Forensic Science International: Genetics, 54. https://doi.org/10.1016/j.fsigen.2021.102551

[12] Schwabenlander, M. D., Bartz, J. C., Carstensen, M., Fameli, A., Glaser, L., Larsen, R. J., Li, M., Shoemaker, R. L., Rowden, G., Stone, S., Walter, W. D., Wolf, T. M., & Larsen, P. A. (2024). Prion forensics: a multidisciplinary approach to investigate CWD at an illegal deer carcass disposal site. Prion, 18(1), 72–86. https://doi.org/10.1080/19336896.2024.2343298

Comments 2: I also have similar concern regarding the attribution of this death to anthropogenic causes. Of course, burns due to forest fire or dog bites are recognizable by experienced vets working in willdife rescue centers but if you have photos to illustrate this, it would significantly improve the manuscript. Forensic sciences are very dependent on necropsy reports and photos and I believe some context regarding the situation of fires in the region (because fires can have human or natural causes) as well as the presence of dogs (because bites could come from other carnivores) is necessary. Basically, authors should answer to the question "why do you believe these lesions represent the anthropogenic threat(s )that this species is facing in the area?". I believe in can be answered in the Discussion, but providing some context first on Introduction or on a first methodology subchapter is also a possibility. 

Response 2: Thank you for this comment and for giving us the possibility to clarify this relevant information. According to the Chilean Government, 98% of the fires were caused by human activity, 2% could not be determined, and none were caused by natural factors, which are uncommon in this country. On the other hand, the wounds in the lumbosacral region were attributed to dog bites because this was the reason for admission provided by the rescuers who transported the animal to the Wildlife Rehabilitation Center. In this sense, we modified the Materials and Method

Lines 76-82: The forest fires that occurred in south-central Chile during the summer of 2023 were primarily ascribed to anthropogenic factors. Official data from the Chilean government indicates that none of the fires recorded during the 2022–2023 season were caused by natural factors [16, 17]. In this context, Wildlife Rehabilitation Center at the University of Concepcion received an adult male southern pudu from Florida, Biobio region, Chile. The patient had wounds in the lumbosacral area attributed to dog bites (as declared by their rescuers) and burned right hind limb, both of which showed signs of infection (Figure S1).”

Additionally, although the wounds are consistent with dog bites, we have placed a sentence in the discussion that a limitation of this study is the lack of confirmation that the bites correspond to a domestic dog or other wild canid.

Lines 282-284: One limitation of this study regarding the wounds in the lumbosacral region is that we could not confirm whether they were caused by attacks from domestic dogs, as reported by the rescuers, or by other wild canids, such as wild foxes.

Photos of the patient's injuries are available in the supplementary Figure S1.

Additional references:

[16] United Nations in Chile. (2023). Chile: Incendios forestales, 2023 - Sistema de Naciones Unidas, Reporte de Situación No. 5 (Al 30 de marzo de 2023) - Chile | ReliefWeb. https://reliefweb.int/report/chile/chile-incendios-forestales-2023-sistema-de-naciones-unidas-reporte-de-situacion-no-5-al-30-de-marzo-de-2023

[17] Gobierno de Chile. (2023). Balance temporada de incendios 2022-2023: 431 mil hectáreas afectadas y 2.369 brigadistas movilizados - Gob.cl. https://www.gob.cl/noticias/balance-temporada-de-incendios-2022-2023-431-mil-hectareas-afectadas-y-2369-brigadistas-movilizados/

Reviewer 2 Report

Comments and Suggestions for Authors

This study isolated bacterial strains and performed whole genome sequencing of key pathogens to characterize features impacting both human hosts and natural habitats. This work provides a valuable reference for future research.

To strengthen the manuscript, the following revisions are recommended:

  1. Clarify preliminary strain identification: While WGS was ultimately employed, the initial methods for species-level identification of bacterial isolates prior to WGS should be explicitly stated. How were the pathogens initially identified (biochemical tests, 16S rRNA sequencing) before WGS?
  2. The Discussion should explicitly state the theoretical basis for concentrating on E.coli ST224. Explain why E. coli ST224 was chosen for detailed analysis over other isolates.

  3. Address whether other isolated strains have potential pathogenic roles, even if not analyzed further.

  4. Replace repeated instances of "whole-genome sequencing" with "WGS" after the first full mention (in Abstract, Introduction…).

  5. Line 131 (Section 2.4): Add a reference for the clonality analysis method. Schürch et al. (2018) should be cited here and included in the bibliography, as it is the basis for SNP-based clonality interpretation.

Author Response

1. Summary

We sincerely thank the reviewers for their valuable time, thoughtful comments, and constructive suggestions, which have greatly improved the quality of our manuscript. Please find below our detailed responses to each comment. All corresponding changes have been incorporated into the revised manuscript and are highlighted using track changes.

2. Questions for General Evaluation

Reviewer’s Evaluation

Response and Revisions

Does the introduction provide sufficient background and include all relevant references?

Yes

.

Are all the cited references relevant to the research?

Yes

Is the research design appropriate?

Yes

Are the methods adequately described?

Can be improved

We respond point by point below.

Are the results clearly presented?

Yes

Are the conclusions supported by the results?

Can be improved

We respond point by point below.

3. Point-by-point response to Comments and Suggestions for Authors

Comments 1: Clarify preliminary strain identification: While WGS was ultimately employed, the initial methods for species-level identification of bacterial isolates prior to WGS should be explicitly stated. How were the pathogens initially identified (biochemical tests, 16S rRNA sequencing) before WGS?

Response 1: Thank you for pointing this out. We performed preliminary strain identification analyses through API 20ETM or API+ 20NETM systems (BioMérieux, France) as mentioned in 2.2. Isolation and bacterial identification subsection. We add the results of this preliminary identification in Table 1 of phenotypic characteristics of the isolates.

The materials and methods section was also modified.

Lines 97-101: An oxidase test for the detection of cytochrome oxidase in microorganisms was performed by using Bactident® (Sigma-Aldrich, USA), according to manufacturer's instructions. Preliminary bacterial identification was performed through API® 20ETM (for oxidase-negative isolates) or API® 20NETM (for oxidase-positive isolates) systems (BioMérieux, France).

Comments 2: The Discussion should explicitly state the theoretical basis for concentrating on E.coli ST224. Explain why E. coli ST224 was chosen for detailed analysis over other isolates

Response 2: Thank you very much for this observation. We have placed a paragraph in the Discussion explaining why detailed genomic analyses were performed for the E. coli ST224 strains.

Lines 245-250: Detailed genomic analysis was prioritized for the two E. coli ST224 isolates to confirm clonal dissemination from a wound to intracardiac blood, indicating progression to septicemia. This clone also carried an extensive set of virulence factors and critical resistance determinants, features not observed in K. oxytoca ST145 (MVL-12-23) or A. baumannii ST1365 (MVL-13-23), which were restricted to localized lesions without evidence of systemic spread.

Comments 3: Address whether other isolated strains have potential pathogenic roles, even if not analyzed further.

Response 3: We Agree. We modified lines 297-301 in Discussion to emphasize this point: “Both K. oxytoca ST145 and A. baumannii ST1365 are recognized opportunistic pathogens with the potential to cause disease in humans and animals. In this case, they were associated only with localized lesions and showed no evidence of systemic dissemination, suggesting that their role was limited to secondary infections rather than primary drivers of the fatal outcome.”

Comments 4: Replace repeated instances of "whole-genome sequencing" with "WGS" after the first full mention (in Abstract, Introduction…).

Response 4: Thank you for pointing this out. We modified the text as recommended.

Comments 5: Line 131 (Section 2.4): Add a reference for the clonality analysis method. Schürch et al. (2018) should be cited here and included in the bibliography, as it is the basis for SNP-based clonality interpretation

Response 5: Thank you for bringing this to our attention. We have made the necessary corrections, including accurate reference:

Lines 141-145: To determine clonality among strains, the clade including our E. coli ST224 strains was subjected to single-nucleotide polymorphism (SNP) analysis using the CSI Phylogeny 1.4 platform of the Center for Genomic Epidemiology, and clonality was interpreted with thresholds following previously described guidelines [20].

Reviewer 3 Report

Comments and Suggestions for Authors

The manuscript “Genomic investigation of bacterial co-infection in southern pudu (Pudu puda) with fatal outcome: application of forensic microbiology in wildlife impacted by anthropogenic disasters” by Valentina Aravena-Ramírez and coauthors conducted a microbiological and genomic study on isolated strains of Escherichia coli, Klebsiella oxytoca, and Acinetobacter baumannii. This study is important given the lack of similar reports in the Pudu genus. However, there are some aspects that need to be clarified or reanalyzed to establish more substantiated inferences.

  • Lines 58-59: Write all scientific names in italics.
  • Section 2.1: Authors should provide further details of the necropsy and the rationale for taking an intracardiac sample. Any signs of endocarditis, pericarditis, etc.?
  • Lines 114-116: Detail the tool used to identify point mutations. RGI?
  • Section 3.2: Authors should report the results of the assembly, particularly those related to its quality (N50, L50, number of contigs, contig size range, completeness, contamination, deep, etc.). These results may be included at least as supplementary material.
  • Figure 1A. I suggest highlighting (if possible) the strains sequenced during the study. Does any of both strains of the study corresponding to the most pronounced branching? If so, it should be discussed.
  • Lines 231-237: These lines deserve further discussion. While SNP analysis shows high similarity between strains, the virulome results are strikingly contrasting (more than 60 differential genes, Table S3). This may indicate that they were different subpopulations and that only one reached systemic status. On the other hand, it may also be due (and is critical to re-evaluate) to differences in assembly, sequence quality, thresholds in virulome tools, among others. The possibility of having acquired all these genes during the infection process is highly unlikely due to the large number of differential genes (several of them usually chromosomal) and the absence of mobile genetic elements detected (except for IncM1, which is usually associated with resistance genes).
  • Lines 241-243: Did the authors attempt to use the MGE finder tool? In any case, they should declare this literally as a limitation of the study.
  • Lines 276-279: Again, this inference is not supported by the results obtained (virulome).

Author Response

1. Summary

We sincerely thank the reviewers for their valuable time, thoughtful comments, and constructive suggestions, which have greatly improved the quality of our manuscript. Please find below our detailed responses to each comment. All corresponding changes have been incorporated into the revised manuscript and are highlighted using track changes.

2. Questions for General Evaluation

Reviewer’s Evaluation

Response and Revisions

Does the introduction provide sufficient background and include all relevant references?

Yes

Is the research design appropriate?

Can be improved

We respond point by point below.

Are the methods adequately described?

Can be improved

We respond point by point below.

Are the results clearly presented?

Must be improved

We respond point by point below.

Are the conclusions supported by the results?

Must be improved

We respond point by point below.

Are all figures and tables clear and well-presented?

Can be improved

We respond point by point below.

3. Point-by-point response to Comments and Suggestions for Authors

Comments 1: Lines 58-59: Write all scientific names in italics.

Response 1: Thank you for pointing this out. We modified italics names.

Comments 2: Section 2.1: Authors should provide further details of the necropsy and the rationale for taking an intracardiac sample. Any signs of endocarditis, pericarditis, etc.?

Response 2: Agree. We modified lines 85-87 to emphasize this point: "Considering the medical records of bacterial infections in the patient and the absence of a established cause of death, an intracardiac blood sample was collected for microbiological analysis at necropsy.”

Comments 3: Lines 114-116: Detail the tool used to identify point mutations. RGI?

Response 3: Yes, we used RGI. We changed line 127-129 to emphasize this point. "For all strains, mutations in the quinolone resistance-determining regions (QRDR) were investigated using the Resistance Gene Identifier (RGI) tool from the CARD database (https://card.mcmaster.ca/home)."

Comments 4: Section 3.2: Authors should report the results of the assembly, particularly those related to its quality (N50, L50, number of contigs, contig size range, completeness, contamination, deep, etc.). These results may be included at least as supplementary material.

Response 4: Thank you for your comment. We analyzed the genome quality of the four isolates using two tools. We modified lines 115-117 in Materials and Methods and attached a supplementary table with the details of the analyses.

Lines 113-116: Genomic assembly was performed using the Shovill (Version 1.1.0) with the SKESA assembler and the quality control and characteristics of genomes was performed using the CheckM lineage_wf and Quast (Table S1) tools available in the Galaxy web-based platform (https://usegalaxy.org/).

In addition, the section on supplementary materials was modified.

Lines 321-326: Table S1: Quality control and genomic characteristics of strains isolated from the southern pudu (Pudu puda). Table S2: Epidemiological data of genomic assemblies of Escherichia coli ST224 available in the Enterobase database; Table S3: Matrix of SNP-based phylogeny analysis of closely related genome assemblies of Escherichia coli ST224; Table S4: Virulence genes (virulome) according to the virulence factor class of Klebsiella oxytoca, Escherichia coli, and Acinetobacter baumannii strains iso-lated from southern pudu.

Comments 5: Figure 1A. I suggest highlighting (if possible) the strains sequenced during the study. Does any of both strains of the study corresponding to the most pronounced branching? If so, it should be discussed.

Response 5: Agree. We modified Figure 1A to highlight the two strains corresponding to the southern pudu isolates. On the other hand, the pronounced branching of the clade does not correspond to either of the two strains in this study.

Comments 6: Lines 231-237: These lines deserve further discussion. While SNP analysis shows high similarity between strains, the virulome results are strikingly contrasting (more than 60 differential genes, Table S3). This may indicate that they were different subpopulations and that only one reached systemic status. On the other hand, it may also be due (and is critical to re-evaluate) to differences in assembly, sequence quality, thresholds in virulome tools, among others. The possibility of having acquired all these genes during the infection process is highly unlikely due to the large number of differential genes (several of them usually chromosomal) and the absence of mobile genetic elements detected (except for IncM1, which is usually associated with resistance genes).

Response 6: We appreciate the reviewer’s insightful observation. We conducted an additional quality control assessment of both genomes, confirming <5% contamination and 99.7% completeness (additional technical parameters are provided in Supplementary Table 1), indicating high-quality assemblies. As SNP analysis considers only the core genome, the high clonality observed reflects conservation in shared genomic regions, while accessory genome differences explain the divergent virulomes. The analysis of mobile genetic elements did not reveal features that could account for the >60 differential virulence genes detected, and the absence of elements typically associated with large-scale gene acquisition during infection (except for IncM1, related to resistance genes) supports the hypothesis that the blood isolate could represent an adapted subpopulation within the same lineage. Although we cannot completely rule out the possibility of virulence gene acquisition via bacterial transformation, the lack of direct evidence prevents confirmation of such an event in this case.

The Discussion was modified. Lines 260-271: “Unfortunately, we cannot confirm the specific mobile genetic elements (MGEs) carrying this extra virulome due to the short-read sequencing that does not allow a correct assembly of plasmids or other MGEs [32]. The marked differences in virulome composition between the two E. coli ST224 isolates, despite their clonal relationship based on core genome SNP analysis, suggest the presence of distinct subpopulations within the same lineage [33]. Given the high assembly quality and the absence of mobile genetic elements typically linked to large-scale virulence gene acquisition, it is unlikely that these differences arose solely during the course of infection. However, the possibility of virulence gene acquisition through bacterial transformation cannot be entirely excluded, although no direct evidence supports this event in the present case. These findings highlight the genomic plasticity within pathogenic lineages and the potential coexistence of subpopulations with variable virulence potential in the same host.”

Additional reference:
[33] Pajand, O., Rahimi, H., Darabi, N., Roudi, S., Ghassemi, K., Aarestrup, F. M., & Leekitcharoenphon, P. (2021). Arrangements of Mobile Genetic Elements among Virotype E Subpopulation of Escherichia coli Sequence Type 131 Strains with High Antimicrobial Resistance and Virulence Gene Content. MSphere, 6(4). https://doi.org/10.1128/mSphere.00550-21

Comments 7: Lines 241-243: Did the authors attempt to use the MGE finder tool? In any case, they should declare this literally as a limitation of the study.

Response 7: After considering your comment, we tested the results using the MGE Finder tool. However, it's important to note that this tool relies on a different database, Virulence Finder, compared to the VFDB used in our research. Despite this, we found no detection of mobile genetic elements associated with the newly acquired virulence factors. As you suggested, we further elaborated on the limitations of our study, as mentioned in your previous comment. Thank you anyway for recommending this tool.

Comments 8: Lines 276-279: Again, this inference is not supported by the results obtained (virulome).

Response 8: Thanks for this observation. We rephrased as: This study suggests that a clonally related E. coli ST224, initially isolated from a wound infection, could cause fatal septicemia in the southern pudu” (lines 310-311).

Round 2

Reviewer 1 Report

Comments and Suggestions for Authors

Authors have addressed my comments and suggestions. Therefore, I have no additional comments to this review

Reviewer 3 Report

Comments and Suggestions for Authors

The authors have responded appropriately to all the comments made in the previous round of reviews.